# Non-invasive recording from the human olfactory bulb

Behzad Iravani [1]*, Artin Arshamian[1,2], Kathrin Ohla [3,4], Donald A. Wilson[5,6] & Johan N. Lundström [1,4,7,8]*

Current non-invasive neuroimaging methods can assess neural activity in all areas of the human brain but the olfactory bulb (OB). The OB has been suggested to fulfill a role comparable to that of V1 and the thalamus in the visual system and have been closely linked to a wide range of olfactory tasks and neuropathologies. Here we present a method for non-invasive recording of signals from the human OB with millisecond precision. We demonstrate that signals obtained via recordings from EEG electrodes at the nasal bridge represent responses from the human olfactory bulb - recordings we term Electrobulbogram (EBG). The EBG will aid future olfactory-related translational work but can also potentially be implemented as an everyday clinical tool to detect pathology-related changes in human central olfactory processing in neurodegenerative diseases. In conclusion, the EBG is localized to the OB, is reliable, and follows response patterns demonstrated in non-human animal models.

[1] Department of Clinical Neuroscience, Karolinska Institutet, 17177 Stockholm, Sweden. [2] Department of Psychology, Stockholm University, 10691 Stockholm, Sweden. [3] Research Center Jülich, Institute of Neuroscience and Medicine (INM-3), Cognitive Neuroscience, Jülich, Germany. [4] Monell Chemical Senses Center, Philadelphia, PA 19104, USA. [5] Nathan Kline Institute for Psychiatric Research, Orangeburg, NY 10962, USA. [6] Department of Child and Adolescent Psychiatry, New York University Langone Medical School, New York, NY 10016, USA. [7] Department of Psychology, University of Pennsylvania, Philadelphia, PA 19104, USA. [8] Stockholm University Brain Imaging Centre, Stockholm University, 10691 Stockholm, Sweden. *email: Behzad.iravani@ki.se; johan.lundstrom@ki.se

Measures of neural processing can be obtained using non-invasive methods from all areas of the human brain but one, the olfactory bulb (OB). The OB is the critical first central processing stage of the olfactory system, intimately involved in processing of an ever-increasing list of olfactory tasks: odor discrimination, concentration-invariant odor recognition, odor segmentation, and odor pattern recognition[1], to mention but a few. Moreover, recent studies demonstrate that the role of the OB is not limited to the olfactory system, but that it impacts many brain functions[2,3]. Within the olfactory system, the OB has been suggested to fulfill a role comparable to both V1[4] and the thalamus in the visual system[5]. Critically, all our knowledge about the OB comes from animal studies. In rodents the relative size of the OB compared to the rest of the brain is very large[6] and as such, it is not surprising that the OB is one of the most well-studied brain areas in the mammalian brain.

The OB is also linked to several disabling neurodegenerative diseases[7] where a strong link to Parkinson's disease stands out[8]. The OB is the very first cerebral area of insult in Parkinson's disease[9] which explains why behavioral olfactory disturbances commonly precede the characteristic motor symptoms defining the disease by several years[10] and why early occurrence of olfactory dysfunction is more prevalent (~91%) than motor problems (~75%)[8,11]. Thus, the development of a non-invasive method to assess OB processing in the awake human is a necessary and important step to fully understand the neural mechanisms of human olfactory processing in both health and disease.

The only published data of human OB odor responses dates back fifty years and was obtained from electrodes placed directly on the human OB during intracranial surgery[12]. Attempts to acquire neural signals from the human OB using functional neuroimaging have failed either due to poor spatial resolution of the method (positron emission tomography; PET) or, in the case of functional magnetic resonance imaging (fMRI), due to the OB's proximity to the sinuses where the cavity creates susceptibility artifacts and reduced signal strength in the OB area[13]. Electroencephalogram (EEG) signals do not suffer from interferences from the sinuses and recordings in rabbits demonstrate that OB signals can be obtained from scalp electrodes placed above the OB[14,15]. However, until now, no attempts have been made to demonstrate non-invasive recordings of OB function in humans using EEG.

Odor-dependent EEG recordings in humans have, by tradition, used low-pass filters at around 30 Hz[16], based partly—on the now disputed assumption—that most human perceptual processes occur in lower frequency bands, and on the observation that human cortical processing of odors mainly operates at around 5 Hz[17]. In sharp contrast, odor processing within the rodent OB has been demonstrated to produce both beta and gamma oscillations[18]. However, when centrifugal input to the OB is eliminated, only gamma oscillations remain[19,20]. Given that gamma and gamma-like oscillations in the OB have been related to odor processing in a range of species[3,21,22] and gamma-band responses have been observed in the only study to date where intracranial recordings from the human OB have been collected[12], we hypothesized that non-invasive signals from the OB, a so-called electrobulbogram (EBG), should be detectable within the gamma-band range. Specifically, this activation should occur within 100–200 ms after odor onset based on the temporal limits given by the biology of the olfactory system (see Supplementary Note 1) and past studies demonstrating that down-stream areas are activated shortly before 300 ms post odor onset[23,24].

To this end, we are addressing the hypothesis that signals from the human OB can be assessed from the scalp using micro-amplified EEG using a four-stage approach. First, we will optimize the electrode placement by simulating how a potential signal would be manifested on the scalp. Second, we will determine whether we can observe an EBG signal on the sensor level that on the source level is located to the OB, with subsequent assessment of reliability of the obtained measure. Third, we will demonstrate that while participants after long odor exposure perceptually habituate, the EBG signal is insensitive to odor habituation. This is a hallmark neural signature of the OB commonly reported in animal models[25]. Finally, using a human lesion-type model—i.e., an individual born without bilateral OBs—we will determine whether absence of OBs abolishes the EBG signal.

## Results

**Determining and localizing the EBG.** We first assessed optimal electrode positions by performing a lead-field simulation where bilateral dipoles where placed in the OB of an anatomical head model (Fig. 1b). Optimal electrode position for signal acquisition was determined on each side of the nasal bridge, just above the eyebrows; standard EEG scalp recording electrode placement charts do not commonly place electrodes there. In Study 1, we therefore placed four micro-amplified EEG electrodes (ActiveTwo, BioSemi, Amsterdam, The Netherlands), two on each side of the nasal bridge (Fig. 1c) to capture the dipole spread and to reduce potential influence of artifacts from single electrodes. Analyses (Fig. 1a, d–f—see the "Methods" section for details) were based on averaged responses to 1 s odor or clean air presentations, presented by a computer-controlled olfactometer[26]. Spectral density of the signal was time-locked to stimulus onset, assessed and adjusted by a photoionization detector[27] and averaged across the four electrodes and trials to optimize signal-to-noise ratio.

Stimuli were triggered shortly after the nadir of the sniff cycle to optimize odor stimulus perception and to eliminate sniff-cycle dependent effects. We therefore first determined whether the motor task of sniffing produced any signal within the designated time and frequency band at the sensor level. To this end, we assessed sniff onset-related responses in the time-frequency map (TFR) within the clean air only condition (Air). Sniff-related activity was indicated in the lower frequency range (~38–45 Hz) just prior to, and around, odor onset (Fig. 2a); however, this sniff-related activity did not differ from baseline in the time-frequency window of interest latter obtained in the Odor versus Air contrast where sniff onset-related activation is cancel out (Fig. 2b; Monte Carlo permutation test with 1000 permutations). We then determined the TFR for odor trials within the designated time and frequency band. To exclude contamination by sniffing and other motor-related artifacts that were not observable, we contrasted the Odor against the Air condition. An odor event-related synchronization (OERS) was observed in the gamma band (~55–65 Hz) around ~100–150 ms post stimulus (Fig. 2d). Subsequent permutation testing (1000 permutations) revealed significant differences between Odor vs. Air conditions. To directly determine the direction of the effect, we compared the averaged power within the time/frequency of interest for each condition against their baseline. Power during the Odor condition (Fig. 2f) was significantly larger than during the Air condition (Fig. 2c), $t(28) = 3.62$, $p < 0.01$, CI [0.23, 0.91] as determined by a Student's $t$-test, providing further evidence that the effect is mediated by the presence of an odor, and not de-synchronization during presentation of air (Supplementary Fig. 1). However, as Fig. 2f demonstrate, an EBG response in the time-window of interest was not clearly detected in all individuals and it is our experience that the exact location (time/frequency) will differ slightly between individuals.

Early visual and auditory sensory responses are often characterized by a phase-locked response to stimulus onset[28]. To assess whether we could detect stimulus phase-locking to odor

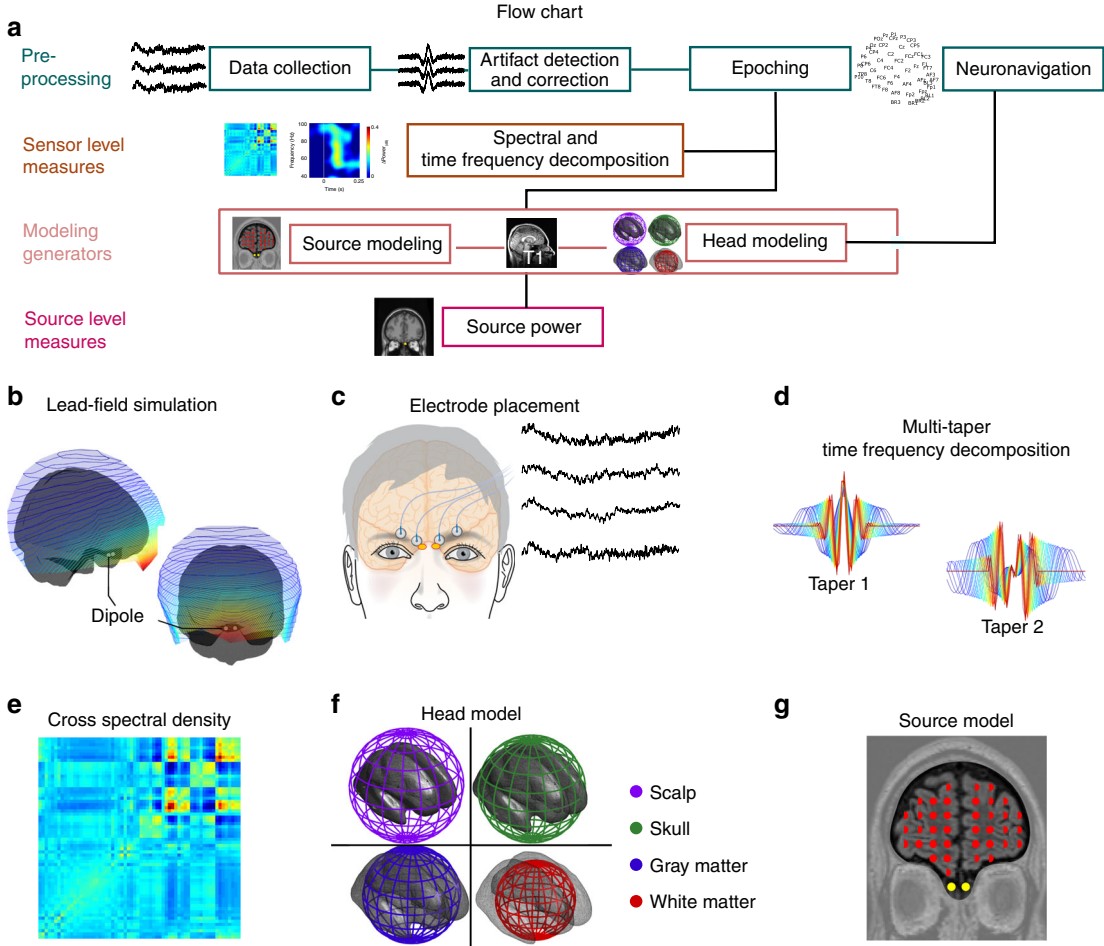

**Fig. 1 Overview of the methodological procedure to extract signal from olfactory bulbs. a** Flowchart of the procedures. **b** A lead-field simulation of olfactory bulb activity projected on the scalp using a symmetrically located dipole in each olfactory bulb (left/right). **c** Electrode placement for the electrobulbogram (EBG) on the forehead and exemplary recordings. **d** Multi-taper time-frequency decomposition using two Slepian tapers. **e** Cross-spectral density between scalp electrodes and EBG channels. **f** Four concentric spheres used to construct the head model. **g** The undetermined source model of every voxel of brain with gray matter probability more than 40%, together with the digitalized sensor position of each individual and head model, were fed into dynamical imaging of coherent source to localize the cortical sources.

onset in the obtained gamma band response in our EBG scalp recordings, we assessed a potential inter-trial phase locking effect in the gamma band and within the same contrast between Odor and Air. There was a change in phase-locked response to the onset of the odor stimuli around the same time point as in our temporal windows of interest used in our analyses (~100 ms), thus bringing additional support to the notion that the EBG signal is an odor-evoked response.

Due to the proximity to the eyes and facial muscles, the EBG measure is artifact sensitive. In Study 1, an average of 52% of all trials was removed from analyses due to artifacts. Thus, to determine the amount of data needed to detect a reliable signal from the EBG with the same statistical power as demonstrated in Study 1, individuals were stepwise added to power analyses. Only seven artifacts free individuals were required to reach full statistical power (Supplementary Fig. 2). From this, we conclude that with the average trial rejection rate, a simple experimental session with one condition would need a minimum of 15 trials to detect a robust EBG signal.

The above detailed power analyses demonstrated that odor stimuli produce a significant EBG signal in the predicted time and frequency domain on the sensor level, and that this was not a function of potential motor and attention-related confounds produced by sniffing. We next asked if the OB is the specific source of this signal. We did this by applying a multi-taper time-frequency decomposition (Fig. 1d) on the signal from all EBG and scalp electrodes in the time/frequency area of interest and localized the signal at single-trial level. Importantly, the individual EEG data was co-registered to a multiple-tissue head model (Fig. 1f) and a source model (Fig. 1g) using a neuronavigation system (Brainsight, Rogue Research, Montreal, Canada) for improved spatial precision. The reconstructed source of the OERS revealed elevated power in the OB, with an 8% increase in power in Odor compared with air condition (Fig. 2g). No other major sources were detected in the time and frequency domain of interest suggesting the OB is, in fact, the underlying source of the EBG signal. To assure that the used source model is the best suited to detect a signal source in the OB, we also assessed the source using eLORETA[29]. Also this competing model localized the source to the OB, albeit with a more dispersed source (Supplementary Fig. 3); probably due to the demonstrated better performance of our initial and main source model for assessments of single sources[30].

The undetermined source model indicated the OB as the underlying source of the EBG but this does not directly compare competing solutions. To directly compare different hypothetical potential signal sources, we used a constrained source-model (guided dipole placement; Supplementary Fig. 4a) to compare the

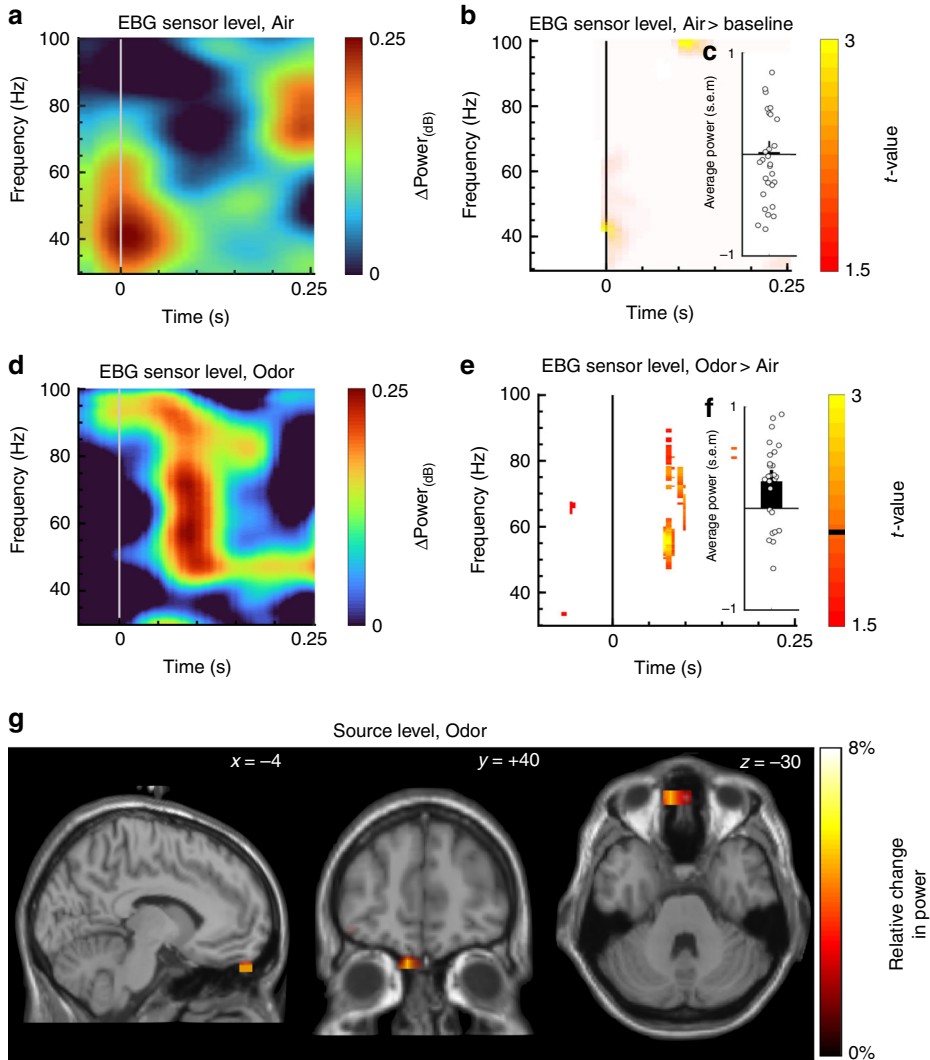

**Fig. 2 Localization of odor-evoked response in sensor and source space (*n* = 29). a** Sensor time-frequency decomposition of difference in power for Air vs. Baseline condition for the EBG electrodes. **b** T-statistics derived from 1000 Monte Carlo permutations demonstrating no change in power for inhalation of Air only condition for the EBG electrodes. **c** Averaged power change for Air across 100–125 ms with standard error of the mean (s.e.m). Circles show individual values. **d** Sensor time-frequency decomposition for Odor against Air conditions. **e** T-statistics derived from 1000 Monte Carlo permutations contrasting Odor with Air conditions ($p < .01$). Orange color marks significant change in power for Odor against Air and the black horizontal line on the color-bar marks the threshold for displayed *t*-values. **f** Averaged power change for Odor condition across 100–125 ms with s.e.m. Circles show individual values. **g** Reconstructed sources of the olfactory evoked synchronization indicating olfactory bulb as the source. Color bars denote relative change in power and *x, y, z* coordinates in figures indicate coordinates of slice in Talairach space according to the MNI stereotactile reference system.

OB, the anterior piriform cortex, the medial orbitofrontal cortex, and, as a non-olfactory control, the primary auditory cortex. The OB solution explained more than twice the amount of the total variance of the signal source space parameters than did dipole solutions in piriform-, orbitofrontal-, and auditory cortex (Supplementary Fig. 4b).

**Reliability and precision of the EBG**. Having established the EBG measure, we next determined its reliability and precision by comparing the EBG in the same individuals across repeated testing sessions spanning multiple days. In Study 2, participants completed three identical testing sessions that were at least one day and at most one month apart. The EBG was acquired using the above described method and analysis focused on the same time and frequency window of interest. First, to determine test–retest reliability, we assessed both intra-class correlation [ICC(2, k)], a measure of agreement[31], as well as pair-wise similarities (i.e., correlation coefficient) between gamma-band

power from both sessions. The ICC(2, k) showed agreement between measurements ($r_i = 0.47$) and subsequent F-test showed that the agreement was statistically significant, $F(2, 26.65) = 3.99$ $p < .03$, indicating a low spread among individuals' EBG values and therefore high agreement[31]. Test–retest correlations ranged between $r = 0.76$ to $r = 0.81$ (Fig. 3a), thereby indicating high test–retest reliability.

Although test–retest correlation is a widely used measure of reliability, the magnitude of a correlation is, to some degree, dependent on the amount of true variability among participants that is in turn dependent on within-participant homogeneity. So, to assess the precision of the EBG measure, we also assessed the mean effect size and the standard error of the mean (s.e.m), an estimate of the standard deviation of the single-trial EBG across an infinite number of sessions. The mean effect size across the three sessions demonstrated a medium effect (Cohen's $d = 0.44$, Fig. 3b) and the s.e.m value across the three sessions (±0.067), compared to a mean power of 0.75, indicate that the EBG measure has good

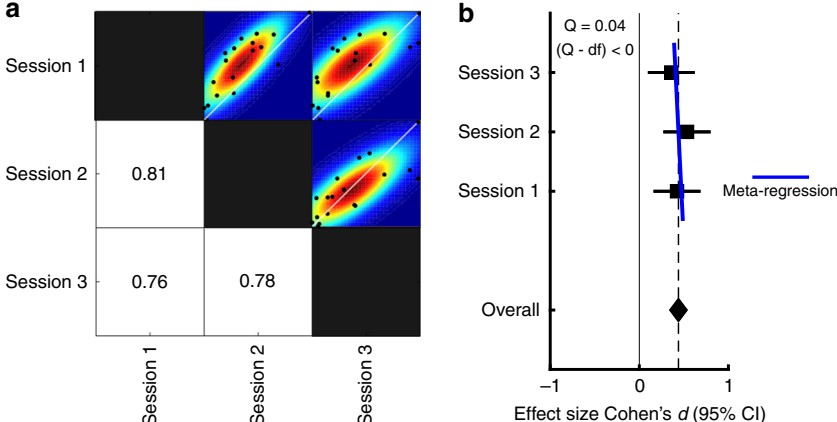

**Fig. 3 Test–retest reliability and dispersion rate of the EBG (n = 18). a** Pairwise correlation matrix across the three sessions. Values indicate bivariate Pearson correlation coefficients and black dots within scatter plots shows individual values for each comparison. Colors indicate mean dispersion with colors indicating smoothed underlying distribution based on bootstrapping of the test data. **b** Effect size and 95% confidence interval for EBG detection within each testing session ($CI_{session1} = [0.16, 0.68]$, $CI_{session2} = [0.27, 0.80]$ and $CI_{session3} = [0.10, 0.62]$). Overall effect showed medium effect size (Cohen's $d = 0.44$) and meta-regression showed insignificant dispersion among the three sessions.

precision. Finally, we assessed dispersion rate using a within experiment meta-regression estimate ($Q$). The dispersion rate indicates whether the distribution is squeezed or stretched compared to an ideal distribution. Assessing the dispersion rate of the three sessions, as determined by help of meta-regression, we found a $Q$ value of 0.04 that is smaller than the experimental degrees of freedom (2) and indicate that the EBG measure has a low dispersion rate (Fig. 3b). Taken together, these data suggest that the EBG measure is both reliable and precise.

**Validating the EBG**. The signal source analyses (Fig. 2g) support the conclusion that the EBG signal originates from the OB. However, the signal source solution is merely the most likely given the acquired data, and not a validation of the method per se. Because no established measure of signal from the human OB exists short of direct and invasive recording from the OB—a measure that is uniquely difficult to obtain due to the ethical dilemma of placing intracranial electrodes that are not strictly needed from a clinical perspective—validation of the measure needs to be indirect. We therefore assessed whether the EBG signal displayed a hallmark signature demonstrated in OB data obtained in several non-human animal models, namely insensitivity to habituation. Importantly, the piriform cortex is known to demonstrate a rapid habituation to repeated or prolonged odor exposure resulting in a clearly diminished neural signal[25,32]. This habituation can be clearly observed in ERPs of the scalp where a short inter trial interval between odor stimuli greatly reduce the signal. In contrast, the signal generated by the OB shows reduced sensitivity to habituation: even after repeated exposure, the OB in rats displays only a minimal reduction in odor-evoked activity[25,33]. Thus, a lack of a significant modulation after rapid, repeated odor presentation would suggest the OB as a primary origin whereas a marked decline of the EBG would indicate that the signal has a major cortical source.

In Study 3, we determined the effect of odor habituation on the EBG response from rapid repetition of odor exposures with long duration, a paradigm that is known to introduce fast and sustained odor habituation[34]. We measured responses from EBG electrodes as well as scalp EEG electrodes. After each trial, participants rated the perceived intensity of the odor on a 10 step computerized visual analogue scale. We first assessed whether our experimental paradigm rendered perceptual odor habituation. As expected, participants experienced a rapid decline in perceived intensity of

the odor on repeated exposure (Supplementary Fig. 5). We then assessed whether the EBG signal demonstrated a similar decline or whether the signal is uncoupled from the perceived intensity of the odor. As predicted by the hypothesis that the EBG signal originates from the OB, a mixed effect model (with trials as fixed effect and subjects as random intercepts) showed no significant slope in OERS power as a function of trial (Fig. 4a). Furthermore, to reduce variability and increase the chance of detecting a potential change, we split the session into two halves (i.e., first half and second half of the session), and statistically tested for potential significant difference between early and late trials in power by 1000 permutations. Although a small nominal decrease in power in the area of interest was observed, there was no statistical change in power between early and late trials (Fig. 4b). We next assessed habituation effects for the scalp ERP signal were we hypothesized that the effect would be large. An established scalp ERP correlate of perceived odor intensity is the difference (delta) in amplitudes between the N1 and P2/3 ERP components over the parietal cortex[35]. The N1-P2/3 difference in power over parietal areas (Pz scalp electrode) demonstrated a characteristic habituation slope with initial large responses that subsequently progressed over trials towards zero (Supplementary Fig. 6). Specifically, the linear trend (linear mixed model) of the effect demonstrated a significant slope across trials, as assessed by a $t$-test, $t(971) = -3.15$, $p < 0.002$, CI $[-0.010, -0.002]$ (Fig. 4c). Together, the results show that the EBG signal possesses the hallmark signature of insensitivity towards odor habituation.

**Validating the EBG response with a human lesion-type model.** Finally, we assessed whether unknown non-olfactory related factors might mediate the EGB response seen in Study 1–3. Although unlikely given the consistency of the EBG signal across experimental conditions, there is a possibility that the EBG signal is mediated by some spurious effect that our experimental designs cannot account for, such as a systematic imbalance in attentional load, task-demands, sniff-related motor activity, micro saccades etc. Therefore, in Study 4, we ruled out these factors by applying the technique to a human lesion-like model by testing whether an EBG signal would emerge when there is no bulb to produce it. We did this by testing one individual with isolated idiopathic congenital anosmia (ICA), i.e., born without the sense of smell. Critically, this individual was without bilateral OBs but otherwise healthy. A magnetic resonance image examination using an OB

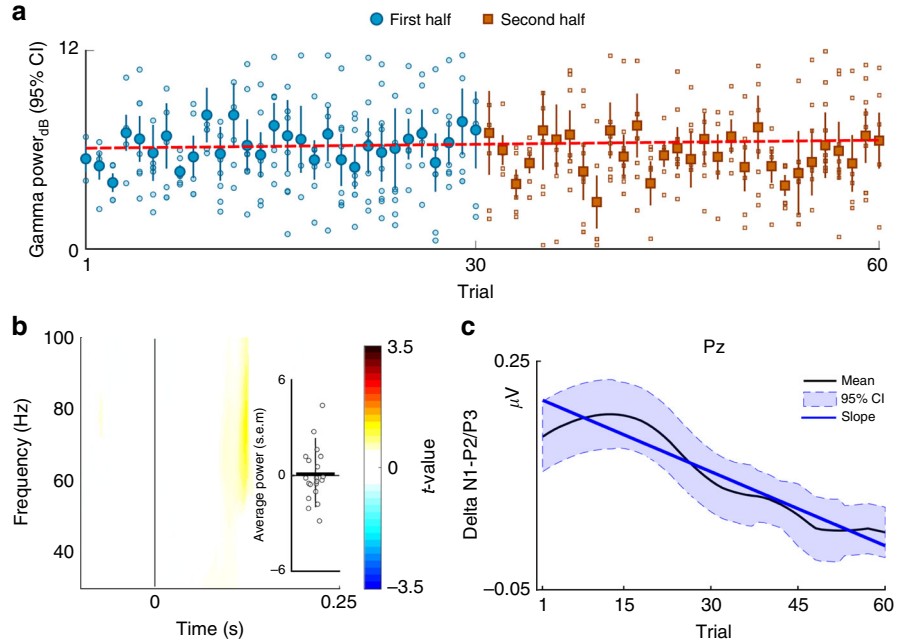

**Fig. 4 Lack of habituation of the EBG measure ($n = 21$). a** Linear mixed model demonstrate that an odor habitation paradigm produce no significant change in power of the olfactory evoked synchronization across trials tested by a one sample $t$-test, slope = 0.008, $t(437) = 1.58$ $p > 0.11$ CI = [−0.002, 0.02]). Blue circles in figure represent the mean of each trial for first half and rust colored cubes the second half of the total number of trials. Error bar show 95% confidence interval. Unfilled circles and cubes represent individual values and dashed line indicate the slope as a function of trials. **b** T-statistics with 1000 Monte Carlo permutations demonstrating no significant change in power ($p > .05$). Positive values indicate larger signal for early trials. Error bar show 95% confidence interval and circles indicate individual values. **c** Pz potentials were band-pass filtered and the local minima and maxima within the time intervals of interest were detected as the peak of N1 and P2/3, respectively. Black curve shows the mean N1-P2/3 peak-to-peak responses in the olfactory event-related potentials (ERPs) over the Pz electrode as a function of trials and shaded blue area shows 95 % confidence interval. The mean curve is smoothed for presentation purposes. Blue line represent the slope revealed by the linear model and indicate a decrease in amplitude difference between N1-P2/P3 ERP components at electrode Pz across trials.

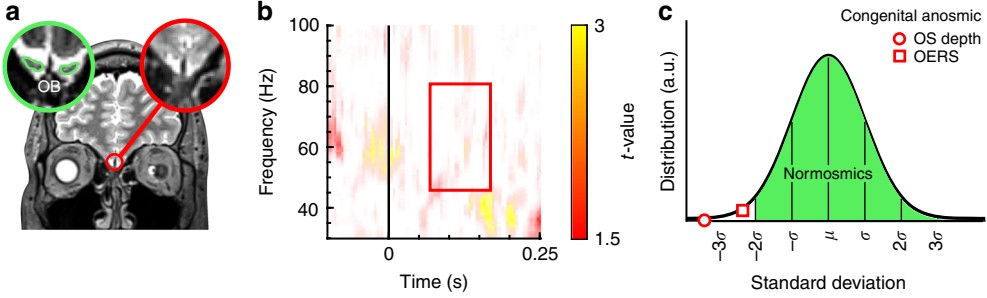

**Fig. 5 Lack of EBG from one individual ($n = 1$) with isolated congenital anosmia (ICA) missing both OBs. a** Coronal view of T2-weighted image of the brain of an individual with ICA, lacking bilateral OB (marked with red). In the left corner is an example of OB from a healthy individual using identical MR examination with the green outline in the green circle delineating the OB. **b** Monte Carlo permutation test with 1000 permutations demonstrating no change in the OERS signal between Odor vs. Air. **c** Distribution of EBG power for the normosmic cohort from Study 1 represented by the green area, the observed sulcus depth (OS) and OB power (OERS) of the individual with isolated congenital anosmia is represented by an open red circle and square, respectively.

sensitive image sequence indicated a complete absence of the OB in both hemispheres (Fig. 5a).

Although there is no definitive test that can distinguish between acquired anosmia, bulb degeneration at very young age, and ICA due to congenital absence of OB, recent studies have reported that an olfactory sulcus depth of less than 8 mm is much more prevalent in ICA patients compared to healthy controls[36]. The ICA subject tested in Study 4 had a mean olfactory depth of 1.12 mm; a value more than 3SD away from an age-comparable control population (Fig. 5c) and, as expected from an individual with anosmia, performed at chance level in a standardized olfactory identification test.

Using an identical experimental protocol as Study 1, we demonstrate that the ICA subject did not exhibit an EBG response following odor stimulation. Specifically, within the time and frequency window of interest, no significant EBG signal was observed for Odors compared with Air condition, Fig. 5b. Visually assessing a single participant's TRF result is inherently difficult due to its noisy structure. We therefore compared the strength of the signal in the time-window of interest to that of all participants in Study 1. As predicted, the mean EBG signal was 2.5 SD below that of the mean of all participants in Study 1 (Fig. 5c). This is yet further evidence that the EBG is sensitive to OB responsiveness.

## Discussion

Even though the OB is the first and, arguably, a critical processing stage of the olfactory neural network, this is the first non-invasive measure of OB processing proposed in humans. The vast majority of olfactory-related electrophysiological recordings targeting odor perception in non-human animal focus on the OB and these explorations have determined that the OB is an important hub for fundamental neural mechanisms across a wide set of topics, including, but not restricted to, memory, learning, social behavior, and motor function[37]. Whether the OB serves the same important role in humans is not known. Delineating cortex from OB activity using electrophysiological measures is inherently difficult. However, using multiple approaches, we demonstrate that the neural processing within the human OB can be non-invasively and robustly measured with electrodes placed at the base of the nose to obtain an EBG. We show that the measure can be obtained with only four EBG and two reference electrodes. We believe this measure is well isolated to OB due to its early occurrence after stimulus onset. We also found that in both forward and inverse models, the OB is a stronger solution as underlying source to the measures signal than potential sources in either the piriform cortex or orbitofrontal cortex (OFC). The EBG measure requires relatively cheap and off-the-shelf equipment and as such, can be easily implemented even with limited financial or computational resources. This method allows for a direct comparison of future studies with humans and already existing non-human animal data. Moreover, the OB is the neural area of initiation of Parkinson's disease[9] and clear behavioral olfactory disturbances precede the characteristic motor symptoms defining the disease by several years[10]. Because a large portion of the OB needs to be destroyed before significant behavioral reduction in olfactory performance is detected[38], recording of EBG signal could potentially serve as a very early marker of PD.

The EBG appears in the gamma band. It is very likely that other signals indicative of OB processing also appears in the alpha, beta, and theta bands at a later time[3]. However, here we focus on the gamma band due to our aim of producing a measure that is localized to the OB and primarily represents processing of the incoming signal with a lesser focus on centrifugal information. Gamma band processing within the OB seems tightly linked to initial intra-bulb processing with limited to no centrifugal influence[18,19]. Indeed, when centrifugal input to the OB is severed, only gamma oscillations can be detected within the OB in response to odors[20,39] whereas beta oscillations are more likely to be modulated by context of odor associations[40]. Similarly, gamma oscillations in the anterior piriform cortex, the area immediately upstream from the OB, are reduced when gamma oscillation is reduced in the OB[41]. Beta oscillations in the anterior piriform cortex are not, however, affected by manipulating gamma in the OB, thus providing further support that gamma activity within the OB reflects within-bulbar processing and potentially OB output—the target of the EBG measure. That said, a plethora of studies in non-human animals have demonstrated that beta oscillations in the OB are very important for the final odor percept. Future studies should thus use the EGB measure to assess the role of beta and alpha oscillations in the human OB. Moreover, the brief activation in lower gamma frequency at stimuli onset in the Air condition is a potential indication that respiration alone may entrain gamma band OB activity; an intriguing question that should be the aim of future studies specifically addressing sniff-induced OB activation.

Our measure is dependent on several key methodological aspects that are required to enhance the EBG signal-to-noise ratio. First, participants were always tested when they were in a nutrition deprived state. This is because in non-human animals, the OB is decidedly more responsive to odors when the individual is in a hungry compared to a satiated state[42,43]. Past studies have demonstrated that more mitral/tufted (M/T) cells are odor responsive when the animal has not been fed, whereas a significant portion of the M/T cells are inhibited during satiation. The fact that we could not obtain a clear EBG signal in a subset of participants could potentially be explained by poor participant compliance with the fasting requirement. Future studies need to assess this potential confound in a systematic manner within a feeding-controlled environment. Second, in all studies but Study 3, odors were presented synchronized to onset of the inhalation phase of the breathing cycle and without a detectable onset cue. About 50% of all M/T cells in the OB are locked to respiration[18,44] and oscillations in the olfactory system, and beyond[45], seem specifically attuned to the respiration cycle. However, note that respiration-locked oscillations normally occur in the theta band and should not be prominently expressed in the gamma band[44]. Third, odors should not have a clear trigeminal perception. Given the automatic motor response of facial frowning elicited by the trigeminal nerve, a part of the pain system, use of trigeminal odors could potentially mask the EBG response[46]. Finally, the measure is dependent on a temporally reliable olfactometer[26] with precise stimuli onset given the dependence on averaging across trails. Jittered onsets would significantly reduce the sensitivity of the EBG measure.

A measure is only useful if it can produce reliable and consistent values that are relatively stable across similar sessions. The EBG measure produced test–retest $r$-values between 0.76 and 0.81; results that are in the same range as established event-related based olfactory and non-olfactory EEG measures. Test–retest of olfactory-derived scalp ERPs normally produce values between as low as 0.05[47] to as high as 0.81[48], dependent on manipulation. Similarly, test–retest coefficients for auditory and visual ERPs are commonly in the 0.48–0.80 range[49]. However, given the low number of trials needed, future development of the measure should consider this potential by including synchronization between an automatic online artifact detection and olfactometer triggering where trials are only initiated when no muscle activity is detected.

Only one publication has presented data originating from surface recordings of the human OB. Hughes and colleagues[12] recorded OB responses to odor stimuli and reported, as do we, predominantly gamma band responses to a range of odors. It could be argued that a weakness of our approach was to base our EBG development on information mostly drawn from studies in non-human animal models. Specifically, one should be aware that the basic assumptions underlying Study 3, demonstrating a lack of habituation in the OB, is based on recordings done on anesthetized animals where later studies have demonstrated that odor-induced neural activity in animals in an awake state do not always generalize well to an anesthetized state[50]. Similarly, separating an OB from a signal source in the OFC based on scalp recordings acquired in humans is a non-trivial task due to the proximity between the two locations. We argue that the EBG response originate from the OB rather than the OFC based on three arguments. First, the signal occurs too early to originate from the OFC (see Supplementary Note 1); second, the OB as a source explained more of the total variance of the recorded signal than other probable source solutions; third, there was no clear habituation detected, a defining feature of neural signals in human perceptual cortex. Nonetheless, the only direct signal validation would be simultaneous recordings from the EBG electrodes as well electrodes placed directly on the OB during odor presentation. However, access to direct recording from the human OB is restricted because measures of OB processing in humans are only possible from recordings done from surgically implanted intracranial electrodes in patients undergoing elected resection surgery

for intractable epilepsy where clinical need direct placement. It is our hope that the EBG measure will produce a richer literature on the role the human OB serves in creating an odor percept, and to delineate similarities and differences of odor processing in human and non-human animal models.

In conclusion, the EBG measure is a valid and reliable measure of signals from the human OB. All needed components are commonly available in most neuroscience institutions and clinical establishments with the one exception being availability of a temporally precise olfactometer. It is our hope that the EBG measures will enable detailed investigations into the role of the OB in the human olfactory system. Specifically, the measure allows the exploration of fundamental mechanistic questions, such as what role the human OB plays in processing odor pleasantness, quality coding, and odor fear learning. Moreover, this method will allow further investigation of a wide variety of clinical disorders known to affect olfactory processing, such as neurodegenerative, eating disorders, as well as schizophrenia.

## Methods

**Participants**. In Study 1, 29 individuals participated (age = 27.07 ± 5.30, 18 women); in Study 2, 18 individuals (age = 28.89 ± 4.80, 7 women) participated in three separate testing sessions on different days; in Study 3, 21 individuals participated (age = 29.55 ± 5.59, 11 women); in Study 4, a 27 years old male, otherwise healthy, individual with the diagnosis of isolated congenital anosmia participated. The diagnosis was confirmed by an ENT physician within the Swedish healthcare system and further supported by our own assessments that indicated that he scored at random when his ability to identify, discriminate, and detect odors was assessed with the standardized clinical odor test Sniffin Sticks[51,52]. Moreover, both his parents, as well as himself, reported no recollection of him ever having an odor sensation and T1-weighted and T2-weighted MR images indicated total absence of bilateral OB and having an average olfactory sulcus depth of 1.12 mm, both morphological measures are indicative of congenital anosmia[36]. All other participants had a functional olfactory sense with no history of head trauma leading to unconsciousness, did not use any prescription drugs, were not habitual smokers, and declared themselves as generally healthy. Functional sense of smell was assessed both by verbal confirmation from the participant and a 5-item 4-alternative cued odor identification test comprising odors from the Sniffin Sticks odor identification test[52]. A minimum of three correct answers were required to participate (mean correct over Studies 1–4: 4.5). Given the low rate of functional anosmia in our tested age group and the known chance score, the likelihood of erroneously labeling an individual with anosmia as having a functional sense of smell is about 0.05%. Participants were recruited through the Karolinska Institutet's participant recruiting site and signed informed consent was obtained before participants enrolled in the respective study. A unique set of participants was used for each study. All aspects were approved by the Swedish national ethical permission board, Etikprövningsnämnden (EPN: 2017/2332-31/1).

**Odor stimuli and odor presentation methods**. We used different sets of odors in the studies to demonstrate the generalizability of results. In Study 1, Orange (Sigma Aldrich, # W282510, CAS 8008-57-9), Chocolate (Givaudan, VE00185273), and n-Butanol (Merck, CAS 71-36-3) were diluted to 30%, 15%, and 20%, respectively, in neat diethyl phthalate (99.5% pure, Sigma Aldrich, CAS 84-66-2). In Study 2, we used Linalool (Sigma Aldrich, CAS 78-70-6), Ethyl Butyrate (Sigma Aldrich, CAS 105-54-4), 2-Phenyl-Ethanol (Sigma Aldrich, CAS 60-12-8), 1-Oceten-3-OL (Sigma Aldrich, CAS 3391-86-4), Octanole Acid (Sigma Aldrich, CAS 124-07-2), and Deithyl Disulfide (Sigma Aldrich, CAS 110-81-6) diluted in neat diethyl phthalate to 0.14%, 0.25%, 0.1%, 0.2%, 1%, 0.25%, respectively. In Study 3, we used 1% isopropyl alcohol (99% pure, Fisher Scientific, CAS 67-63-0) diluted in Propylene Glycol (99% pure, Sigma Aldrich, CAS 57-55-6). In Study 4, Chocolate (Givaudan, VE00185273), n-Butanol (Merck, CAS 71-36-3), and 1-Oceten-3-OL were diluted to 15%, 20% and 1%, respectively, in neat diethyl phthalate. All dilution values above are given as volume/volume from neat concentration.

In all studies, odors were delivered birhinally using a computer-controlled olfactometer with a known rise-time (time to reach 90% of max concentration from triggering) of about 200 ms[26] and a total flow-rate of 3 L/min/channel and inserted into an ongoing 0.3 L/min constant flow to avoid tactile sensation of the odor onset. This means that total airflow per nostril was never higher than 1.65 L/min, a flow significantly lower than airflows known to elicit nasal irritation[26].

The olfactory and respiratory system are tightly intertwined. To remove potential effects of respiration from the measure, we used a sniff-triggered design: in Study 1, 2, and 4, all trials were initiated at the onset of inhalation. This was achieved by monitoring the sniff pattern by means of temperature pod attached close to the right nostril sampling at rate of 400 Hz (Powerlab 16/35, ADInstruments, Colorado) and processed in LabChart Pro version 8.1.13. As the individual breathes in, the cold air lower the temperature and as the person

breathes out, warm air elevates the temperature. The change of temperature therefore indicates the respiration cycle. An individual threshold was set to trigger the olfactometer slightly before the nadir of the respiratory cycle to synchronize odor presentation with nasal inspiration. In Study 3, we employed a different strategy to remove the effect of respiration by instructing the participant to breathe through their mouth throughout the study, thus abolishing the sniff cycle, and the odor stimuli were passively presented.

Stimulus triggering and timing was achieved using E-prime 2 (Psychology Software Tools, Pennsylvania). To avoid participants predicting the onset of the trial, a jittered pre-stimulus interval (600~2000 ms) was inserted before each trial. Moreover, to minimize habituation, a long inter-trial-interval (ITI) was initiated after odor offset (14,000 ms), except in Study 3 where habituation was sought. Moreover, to minimize potential redundant disturbances, participants were tested in a sound attenuated recording booth with good ventilation and potential sounds from the olfactometer and odor mixing manifold, which might give away odor onset, was masked with low volume white noise presented via headphones throughout the whole experiment. The volume of noise was adjusted for each individual to keep them comfortable through the full experiment.

**Electrode placement for EBG**. The optimal location of the EBG channels were determined based on simulated lead-field. The scalp lead-field were simulated for two dipoles placed in left and right OB. The left and right OB location were determined on the native space of individual T2-weighted images in ACPC coordinate system and converted to the MNI coordinate system; left OB ($x$ −6, $y$ 30, $z$ −32) and right OB ($x$ 6, $y$ 30, $z$ −32).

The dipoles momentum was assumed to face radially outward and the same head model (i.e., four co-centric spheres) as the main analysis was used to project the lead-field on the scalp level (Fig. 1b). The simulation suggested that the majority of the OB's energy concentrate on the forehead; therefore, optimal placement of the four electrodes were determined to be a curved configuration on the forehead slightly above the eyebrows, bilaterally, in addition to two mastoid electrodes as the reference electrodes (Fig. 1c; mastoid electrodes are not shown in the figure). For detailed implementation of lead-field estimation, please see ref. [53].

**Electroencephalogramy and neuronavigation measurement**. In all studies, the EEG (acquired using either 32 or 64 electrodes, dependent on study) and EBG (acquired using four additional frontal electrodes) signal was sampled at 512 Hz using active electrodes (ActiveTwo, BioSemi, Amsterdam, The Netherland) and band-pass filtered at 0.01–100 Hz during recording within the ActiView software (BioSemi, Amsterdam, The Netherland). Before the actual EEG/EBG recording, electrode offset of each electrode was visually checked and electrodes with offset above 40 mV was adjusted until the offset reached below the accepted threshold value. EEG electrode placement followed the international 10/20 standard in all studies and two mastoids electrodes were used as reference.

In Study 1 and 2, the EEG/EBG recording included 64 EEG scalp electrodes and 4 EBG electrodes. After the attachment of all electrodes, the positions of each electrode in stereotactic space were digitalized using an optical neuro-navigation system (BrainSight, Rogue Research, Montreal, Canada). The digitalization protocol comprised of localizing fiducial landmarks such as the nasion and left/right preauricular as well as the central point of each electrode. These landmarks were next used to co-register each electrode to the standard MNI space. The digitalized electrode positions were later used in the Beamforming algorithm to enable the localization of cortical sources. In Study 3, data were recorded from 32 EEG scalp electrodes and 4 EBG electrodes and Study 4 used 64 EEG scalp electrodes and 4 EBG electrodes.

**Preprocessing of EEG and EBG data**. EEG/EBG signals were preprocessed by epoching data from 500 ms pre-stimulus to 1500 ms post stimulus. Next, data were re-referenced to the average of left and right mastoids electrodes, band-pass filtered at 1 Hz–100 Hz, and line-filtered at electrical frequency. The line filtering was performed with discrete Fourier transform (DFT) filters in which we applied a notch filter to the data to remove power line noise. The notch filter was implemented by fitting a sine and cosine function to the data at power line frequency with subsequent subtraction of the noise component. The epoch length in all analysis was at least 2 s, covering 100–120 cycles of power line noise component and led to sharp spectral of the notch filter. This sharp spectral feature of the notch filter increases the specificity of removing the noise component[54,55]. Furthermore, trials with different types of artifacts (i.e. muscle and eye blinks) were identified with automatic algorithms. Identifying muscle artifacts was performed by band pass filtering the raw data using Butterworth filter order of 8 and Hilbert transformed to extract amplitude values, followed by z-score. Trials with z-value above 6 were identified as trials contaminated by muscle artifact and removed from further analysis.

Trials with eye blinks were identified by band-passing the raw data by Butterworth filter order of 4 and Hilbert transformed to extract amplitude values, followed by z-score. The major concern for EBG signal is eye blinks and eye movements therefore, a lower z-value of 4 was used to increase the detection sensitivity of the algorithm. Trials with value exceeding 4 were removed from further analysis. Finally, a manual inspection was carried out and trials with comparative high variance were removed.

**EBG time-frequency analysis for OERS detection**. Development of power across time and frequency of the EBG channel in the gamma frequency was determined by employing a multi-taper sliding window (range 30–100 Hz with step 0.1 Hz). Power was estimated at each bin using wavelet with two tapers from discrete prolate spheroidal sequences (DPSS). The window length was adjusted to capture three cycles of the signal at each frequency bin. For lower frequencies, we considered a wider window and as the frequencies reaches higher value the window also becomes narrower. A narrower window at the higher frequencies increases the sensitive of the power estimation by implementation of a higher time resolution but also lower frequency resolution. In the gamma band, lower frequency resolution is not a significant confound because the gamma band is considered to be broadband (i.e., 30–100 Hz). Wavelets transformation at each time bin was carried out by two sets of wavelet function that was derived from the two DPSS tapers. To perform the wavelet transformation, the wavelet function had to convolve with the EEG/EBG signal (Fig. 1d). The convolution was implemented in the frequency domain as a multiplication of fast Fourier coefficients of the signal and the wavelets. Next, the estimated power of the epochs was demeaned by normalizing to the average power of the whole epoch and converted to decibel values (dB).

**Localizing the OERS' cortical source**. To localize the cortical source of the detected OERS, EEG/EBG were re-referenced to average of electrodes and spectral density of the signal at the time period of 100–250 ms post-stimulus were estimated using fast Fourier algorithm with central frequency 60 Hz (i.e., the central frequency of the OERS) and taper smoothing parameter 5 Hz, meaning that the range from 55 Hz to 65 Hz were taken into the computation of cross-spectral density for source localization (Fig. 1e). The number of tapers was estimated as the time half bandwidth[55]. Prior to cross-spectral density estimation noisy electrodes were identified by examining the power of power line noise (50 Hz) of the electrodes for given time window, those electrodes with $z$-value more than 3.5 were interpolated using weight average of the adjacent electrodes. Then, the cross-spectral density between pairs of electrodes was derived by multiplying the spectral density of a channel with conjugated spectral density of other channels. To solve the inverse problem on a trial level, a linear transformation of dynamic imaging of coherent sources (DICS) was used[56]. Given the associations among electrodes, a unique configuration of cortical sources can be estimated by DICS that explains the scalp potential. Association among electrodes was measured by coherence derived from cross-spectral density between pairs of channels at the central frequency (i.e. 60 ± 5 HZ). We also assigned the regularization parameter to 10% in order to reduce the effect of nuisance parameter.

Digitalized electrode-positions of each participant were co-registered to the default MNI brain. Co-registration was performed automatically with six parameters affine transformation followed by manual inspection for any misalignment. Subsequently, a head model was created based on a multi-shell spherical head model. Construction of the head model was initiated by tissue segmentation on the default MNI T1-weighted image. The segmentation procedure included scalp, skull, gray matter, and white matter. Next, spherical volume conductors with the conductivity of 0.43, 0.01, 0.33, and 0.14 were assigned to scalp, skull, gray matter, and white matter respectively (Fig. 1f). An underdetermined source model was used in which distributed sources were equally spread over the full brain. The brain was divided into a three dimensional grid, covering the whole brain with at least 10 mm spacing between two points on the grid. We constructed the source model on each grid-point depending on the gray matter probability of that particular point. A dipole was placed on the points with the gray matter probability larger than 40% (Fig. 1g). The DICS algorithm looks for a weighted summation of the scalp electrodes in order to reconstruct the cortical sources on trial level. We used the balanced common filter approach: here, sources for both conditions (i.e. Odor and Air) were concatenated and a common solution for the inverse problem was computed. Therefore, the difference of the cortical sources between two conditions is free from biases originating from different solutions estimated by DICS. Subsequently, Odor trials and Air trials were averaged within individuals. Moreover, to quantitatively investigate the goodness of fit for the inverse model, we used the dipole fitting approach (determined source model) to assess the amount of power each hypothetical sources can explain. Multiple sites of the brain selected including OB, anterior piriform cortex, orbitofrontal cortex, and a non-olfactory related area, primary auditor cortex, as the underlying sources. Two symmetric dipole place in each of these sites bi-hemispherically. Then, the forward problem was solved with the same head model as inverse problem within time frequency of the interest for each scenario and the explained power and error estimated. DICS analysis were carried out in the open source Fieldtrip toolbox[57].

**Statistical analysis**. All statistical analyses were performed within the MATLAB (version 2018a) environment with Signal Processing and Fieldtrip toolboxes. The spectral density of the four EBG electrodes were averaged on the participant level. Then, Monte Carlo permutation tests was used to examine if the power of averaged EBG spectral density was significantly higher in Odor compared with Air on the group level. Non-parametric permutation tests were used to assess statistical significance rather than parametric statistics given the tests ability to assess a sharp null hypothesis (i.e. no difference between conditions), its ability to provide exact control of false positives, because the EBG measure is producing an unknown distribution, and the increase in generalization of obtained results. A 1000 permutations were

performed on the averaged EBG spectral density so that in each permutation, 50% of conditions where shuffled and the difference between Odor and Air calculated by means of two tailed t-test between the actual data and shuffled data. The exact $p$-value was derived by the average number of the times that the actual data is bigger than shuffle data out of 1000 permutations. For purpose of illustration, $t$-maps were smoothed while maintaining the shape of the observation. Standard conservative corrections for multi-comparison could not be employed due to extensive number of test elicited by the high resolution of the spectral density maps. We therefore reduced the risk of false positive results induced by the many statistical tests by replicating the main EBG finding in independent experiments (Study 1, 2, and 3). Test–retest reliability was assessed by bivariate Pearson correlations and interclass correlation determined with ICC(2, k), a measure widely used to quantify the agreement of the target measure (i.e. OERS power) between individuals across different sessions[31]. All tests, when applicable, are two-sided.

**Reporting summary**. Further information on research design is available in the Nature Research Reporting Summary linked to this article.

## Data availability
Data is freely and publicly available at: [https://osf.io/64tdk/?view_only= a544ade42c9443f0a2842a7eebca7138]. A reporting summary for this Article is available as a Supplementary Information file.

## Code availability
Code to reproduce data visualized in Figs. 2 and 3 are available in the above mentioned link in the Data availability section.

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

## Acknowledgements

We thank Christopher Maude, Kristen Gregory, and Martin Schaefer for assistance with data collection, Dr. Asifa Majid for helpful comments on the manuscript, and Kimberly Battista (battistaillustration.com) for the making of Fig. 1c. Funding provided by the National Institute on Deafness and Other Communication Disorders (R21DC016735) as well as the Knut and Alice Wallenberg Foundation (KAW 2018.0152), awarded to J.N.L., A.A. is supported by a grant from the Swedish Research Council (2018–01603).

## Author contributions

Conceptualization, J.N.L.; Investigation, B.I., K.O.; Methodology, J.N.L., B.I., A.A., K.O., D.W.; Analyses, B.I.; Visualization, B.I., A.A.; Writing—Original Draft, J.N.L., B.I., A.A.; Writing—Review and Editing, all authors; Funding Acquisition, J.N.L.

## Competing interests

A patent has been filed to protect the method from unwanted commercialization by third parties. Aspect of manuscript covered in patent application: Experiment 1 within the manuscript. Experiments 2–5 not mentioned/covered.
