## [Peer Review File · Nature Communications]

Reviewers' Comments:

Reviewer #1:

Remarks to the Author:

This very strong paper describes methodology for recording the EEG signal from the human olfactory bulb. Using a clever arrangement of electrodes and analysis tools, the authors convincingly argue that they have isolated the signal from the OB. I have no major concerns and believe that this work represents a significant advance in methodology which will allow registration of the OB signal in both healthy and disease models. Validation of the signal source used several clever methods, including a lack of correlation with perceptual habituation and absence of the signal in subjects without an OB. Note: I am not an expert in human EEG source localization and so cannot comment on the strength of those methods, but the methods themselves are well-described and appear sound to a non-expert.

Minor comment

Figure 2. Please put the time-frequency plots on the same color scale.

Reviewer #2:

Remarks to the Author:

The authors report four studies on the development of a non-invasive technique recording direct neural potentials from nerve fibers of the olfactory bulb.

This is a very interesting technique of great potential value for the community of human olfactory Research.

The manuscript is sound. I honestly struggle following each statistical step in detail, but the results are convincing and documented in a transparent way so that it should be possible to replicate the experiments and analysis.

It was a true pleasure to read this manuscript and I am convinced that this will be of help for the scientific community.

I have no further comments and recommend acceptance.

Reviewer #3:

Remarks to the Author:

This carefully organized project presents important evidence that the function of the olfactory bulb can be assessed with a simple noninvasive EEG measure. There is considerable clinical significance of assessing anosmia in routine practice, such as in detecting early Parkinson's or other dementia. The convergent measures provide evidence of variation in the frontopolar gamma signal that is consistent with the hypothesis that it reflects functional activity in the olfactory bulb.

Although the alternative source for the frontopolar gamma, the olfactory cortex, is considered, and arguments for the specificity of the measure to olfactory bulb are presented, it might make for a stronger manuscript to acknowledge the possibility that the olfactory cortex contributes to the measure. Clinically and experimentally, the response of the olfactory system is clear in both cases, and the attribution to olfactory bulb seems more of academic interest.

Although the simulation suggests how the source in olfactory bulb would propagate, it is not clear that the scalp pattern could separate a similar source in olfactory cortex.

The person without an olfactory bulb is also without input to olfactory cortex.

It is a reasonable argument that greater habituation suggests the bulb, but Wilson 1998 was with anesthetized animals which may not generalize well to the awake state (Rinberg 2006).

A brief statement on the difficulty of separating bulb and cortex in the discussion would make for a more balanced report.

I would also like to see a brief note on the ability to generalize to a clinical measure (not using the complex olfactory delivery). This seems quite feasible and indeed important to allow this important discovery to contribute to olfactory assessment in routine practice.

Minor edit:

Change: average of 52% of all trials were removed

To: average of 52% of all trials was removed

Reviewer #4:

Remarks to the Author:

This study investigated if neuronal activity can be measured from human olfactory bulb (OLB) non-invasively using EEG. To this end, the EEG electrodes were placed in the head by the help of forward modelling of the OLB activity, and was found to reliably record the activity of the OLB. This approach was termed electrobulbogram (EBG). The EBG also followed response patterns demonstrated in non-human animal models. The manuscript brings up unique information and possibilities of recording activity from OLB humans that has not been possible this far. The manuscript is well written and the observations are well validated. Yet, there are many unclarities in the methods and analyses approach taken that should be clarified.

Major

One of the main validation in the study was the observation that ICL patient without OLB does not have gamma oscillatory activity similarly to healthy subjects. Figure 5b shows data for this one ICL patient. These data is difficult to compare with the corresponding data of healthy subjects presented in Figure 2 as this is an average over all subjects. It would be valuable to present single subject TFR data also to healthy participants for a comparison.

The optimal location of the Electrobulbogram (EBG) channels were determined based on simulated leadfield. Please give more insight of what head models were used and how the leadfield was constructed.

Source modelling was performed with DICS, which is a model based on suppression of coherent sources. Bilateral primary sensory areas including OLB are usually highly synchronous with each other and therefore the other of the sources is largely suppressed. Why was the DICS chosen for source-modelling these data as it can't give accurate reflection of neuronal activity in bilateral OLB. It should be shown that other source modelling approaches such as MNE or Loreta can be used to better localize distributed sources would give similar results.

Only oscillatory power changes were analysed. Yet, the early responses are also phase-locked to the stimulus onset. This so called inter-trial phase locking is also known to play a significant role in sensory perception in other modalities. As OLB is in the first stages of olfactory processing, the gamma-band responses would be expected to be also phase-locked to the stimulus onset. This should be established.

In p. 10, N1 was reported to be present over parietal areas. I am confused as in the Method section, it is described that responses were recorded with 4 electrodes. Also the only figure of this is figure showing the decay of the N1 component as function of trials. Please clarify and explain in detail the

measurement and approach chosen. Also show the ERPs as function of time similar to that used for showing the TFR. Overall, I think that the impact of the study would be stronger if the present results from the OLB could be tied within the results and neuronal phenomena found at the later stages of cortical processing recorded with classical EEG recordings. More specifically, if gamma oscillations characterize the processing in OLB, what are the next stages in processing the olfactory response? Perhaps the P50 or N1 responses.

The lesion was suggested to control several physical factors, including respiration. Yet, it is quite clear that respiration and sniffing are intertwined in healthy participants. I am intrigued about how these would be coupled. Respiration rate is ~ 1 Hz while the odor perception was correlated with gamma-band oscillations. Could it be so that the gamma oscillations are cross-frequency coupled to the breathing rate?

Please clarify, how the EEG channels were referenced.

I find the summary confusing and not descriptive of the actual study. First, it does not mention that recordings are obtained from human subjects. Second, the description of EBG is quite misleading and it is used both as technique and result. Please clarify.

The introduction summarizes many prior studies of neuronal activity on OLB. It would be good if the also the seminal studies of the group by Gilles Laurent which reveal the computational functions and operations carried out by gamma oscillations in olfactory processing could be referred to (See for instance Laurent 2002 Nature Neuroscience Nov;3(11):884-95).

Minor

Add minor tic marks to Fig 2 x-and y-axis.

Add the N to the figure legends. The color-scale is difficult to read i.e. the different T-values are impossible to extract. Please change to a scale which gives more resolution for the values in the TFRs.

Response to reviewers: Non-invasive recording from the human olfactory bulb.

We thank the Editor and the four reviewers for their comments on our manuscript. Below, we have addressed all comments in a point-by-point fashion, grouped according to reviewer. In summary, we have changed our figures, added additional analyses (per request by Reviewer #4), and cleared up misunderstandings or areas of confusion over our text as well as expanded on topics requested by Reviewer #3 and #4. In addition, we have also formatted the manuscript to adhere to Nature Communication's formatting demands. All changes within the manuscript itself is highlighted with **yellow marking**. Overall, we think that the manuscript is more informative, with more data, yet presents a more coherent bulk of data than in the first version. Our responses to reviewer comments below are marked with **blue text** below and excerpts from the changed text within the manuscript are marked with *italic text*. We wish to thank all the reviewers for their attention to details and for their kind and informative comments.

Reviewer #1 (Remarks to the Author):

Reviewer: This very strong paper describes methodology for recording the EEG signal from the human olfactory bulb. Using a clever arrangement of electrodes and analysis tools, the authors convincingly argue that they have isolated the signal from the OB. I have no major concerns and believe that this work represents a significant advance in methodology which will allow registration of the OB signal in both healthy and disease models. Validation of the signal source used several clever methods, including a lack of correlation with perceptual habituation and absence of the signal in subjects without an OB. Note: I am not an expert in human EEG source localization and so cannot comment on the strength of those methods, but the methods themselves are well-described and appear sound to a non-expert.

Minor Edit

R1. Minor.Comment.1:

Reviewer: Figure 2. Please put the time-frequency plots on the same color scale.

Reply: We thank Reviewer #1 for their kind assessment of our work. To address their one comment, we improved Figure 2 by changing the color map to a more linearly perceived color gradient (Turbo; <https://ai.googleblog.com/2019/08/turbo-improved-rainbow-colormap-for.html>) and put the two time frequency maps on the same scale. We also added minor ticks to figure to increase clarity (per request by Reviewer #4).

New Figure 2.

Reviewer #2 (Remarks to the Author):

Reviewer: The authors report four studies on the development of a non-invasive technique recording direct neural potentials from nerve fibers of the olfactory bulb. This is a very interesting technique of great potential value for the community of humans olfactory Research. The manuscript is sound. I honestly struggle following each statistical step in detail, but the results are convincing and documented in a transparent way so that it should be possible to replicate the experiments and analysis.

It was a true pleasure to read this manuscript and I am convinced that this will be of help for the scientific community.

I have no further comments and recommend acceptance.

Reply: We thank the review for evaluating our work and for their positive comments.

Reviewer #3 (Remarks to the Author):

Reviewer: This carefully organized project presents important evidence that the function of the olfactory bulb can be assessed with a simple noninvasive EEG measure. There is considerable clinical significance of assessing anosmia in routine practice, such as in detecting early Parkinson's or other dementia. The convergent measures provide evidence of variation in the frontopolar gamma signal that is consistent with the hypothesis that it reflects functional activity in the olfactory bulb.

Although the alternative source for the frontopolar gamma, the olfactory cortex, is considered, and arguments for the specificity of the measure to olfactory bulb are presented, it might make for a stronger manuscript to acknowledge the possibility that the olfactory cortex contributes to the measure. Clinically and experimentally, the response of the olfactory system is clear in both cases, and the attribution to olfactory bulb seems more of academic interest.

R3 Comments 1:

Reviewer: Although the simulation suggests how the source in olfactory bulb would propagate, it is not clear that the scalp pattern could separate a similar source in olfactory cortex. The person without an olfactory bulb is also without input to olfactory cortex.

Reply: The reviewer is accurate in that the individual with isolated congenital anosmia is not a control for a potential confounding signal from other odor-related areas, merely a control for other non-odor-related sources (such as motor response, etc). In respect of separation of sources, the lead field simulation was not intended to indicate signal source location but, rather, to answer the question "if an OB signal can be detected, where is the theoretical best recording position". We have clarified this further in the text by amending the first paragraph of the Result section to include this statement:

"Optimal electrode position for signal acquisition was determined on each side of the nasal bridge, just above the eyebrows; standard EEG scalp recording electrode placement charts do not commonly place electrodes there."

To address the underlying question of separation of sources -- we believe that the timing of the signal (about 100-150ms after odor onset) makes an OFC source much less likely (see Supplementary Data for a calculation of signal transfer time); however, we accept that tem-

poral aspects alone do not render an OFC source impossible given that only two studies have directly assessed time to onset in piriform cortex (about 300ms from odor onset) using intracranial electrodes. In those studies, it is unclear if they corrected for odor onset time (i.e. time between olfactometer trigger signal to odor at the receptors). To assess whether the signal originates from the OB or OFC in a more data driven manner, we forced the source (dipole) in our analyses to be in the OB, the piriform cortex, or the OFC and assessed total power of the signal. The results from this analyses can be found in Supplementary Figure 3B. We found that the OB solution explains more power compared with other regions (double as much), including the orbitofrontal cortex. This data, paired with the early occurrence of the gamma synchronization, makes us argue that it is improbable that orbitofrontal cortex is the underlying source of the EBG signal. However, we have added a discussion on the accuracy of our underlying source result in the Discussion section, namely:

“Similarly, separating an OB from a signal source in the OFC based on scalp recordings acquired in humans is a non-trivial task due to the proximity between the two locations. We argue that the EBG response originate from the OB rather than the OFC based on three arguments. First, the signal occurs too early to originate from the OFC (see Supplementary data); second, the OB as a source explained more of the total variance of the recorded signal than other probable source solutions; third, there was no clear habituation detected, a defining feature of neural signals in human perceptual cortex. Nonetheless, the only direct signal validation would be simultaneous recordings from the EBG electrodes as well electrodes placed directly on the OB during odor presentation. However, access to direct recording from the human OB is restricted because measures of OB processing in humans are only possible from recordings done from surgically implanted intracranial electrodes in patients undergoing elected resection surgery for intractable epilepsy where clinical need direct placement. ”

R3. Comment 2:

Reviewer: It is a reasonable argument that greater habituation suggests the bulb, but Wilson 1998 was with anesthetized animals which may not generalize well to the awake state (Rinberg 2006).

Reply: This is an excellent point that we are now mentioning in the new version of the manuscript. On page 15 in the Discussion, we have added the following text:

“It could be argued that a weakness of our approach was to base our EBG development on information mostly drawn from studies in non-human animal models. Specifically, one should be aware that the basic assumptions underlying Study 3, demonstrating a lack of habituation in the OB, is based on recordings done on anesthetized animals

where later studies have demonstrated that odor-induced neural activity in animals in an awake state do not always generalize well to an anesthetized state (Rinberg et al. 2006)."

R3 Comment 3:

Reviewer: A brief statement on the difficulty of separating bulb and cortex in the discussion would make for a more balanced report.

Reply: This is a fair critique and as indicated above under R3 Comment 1, we have added a section in the Discussion where this is discussed. In addition, we have added discussion on this point in the first section of the Discussion. This now reads:

"Delineating cortex from OB activity using electrophysiological measures is inherently difficult. However, using multiple approaches, we demonstrate that the neural processing within the human olfactory bulb (OB) can be noninvasively and robustly measured with electrodes placed at the base of the nose to obtain an electrobulbogram (EBG). We show that the measure can be obtained with only four EBG and two reference electrodes. We believe this measure is well isolated to OB due to its early occurrence after stimulus onset. We also found that in both forward and inverse models, the OB is a stronger solution as underlying source to the measures signal than potential sources in either the piriform cortex or orbitofrontal cortex (OFC)."

R3 Comments 4:

Reviewer: I would also like to see a brief note on the ability to generalize to a clinical measure (not using the complex olfactory delivery). This seems quite feasible and indeed important to allow this important discovery to contribute to olfactory assessment in routine practice.

Reply: Good point. The brief activation of lower frequency, ~40Hz in Air condition compared to baseline in Figure 2A, could be a potential indication that respiration alone can entrain OB responses with no need of olfactory delivery. We hope to investigate this further in future work. Note, however, given the fact that we do not have any data on a potential clinical usefulness of using a sniff-induced EBG measure, we prefer to not allude to that this can be used as a clinical measure. If the Reviewer and Editor insist on inserting this, we would of course include the information with an added caveat of it being speculative. Nonetheless, we have now added a sentence on p.14 as follows:

"That said, a plethora of studies in non-human animals have demonstrated that beta oscillations in the OB are very important for the final odor percept. Future studies should thus use the EGB measure to assess the role of beta and alpha oscillations in the human olfactory bulb. Moreover, the brief activation in lower gamma frequency at stimuli onset in the Air condition is a potential indication that respiration alone may entrain gamma

band OB activity; an intriguing question that should be the aim of future studies specifically addressing sniff-induced OB activation."

Minor Edit:

R3 Minor Comment 1:

Reviewer: Change: average of 52% of all trials were removed To: average of 52% of all trials was removed

Reply: Thank you for pointing this out. We changed the sentence accordingly in the manuscript.

Reviewer #4 (Remarks to the Author):

Reviewer: This study investigated if neuronal activity can be measured from human olfactory bulb (OLB) non-invasively using EEG. To this end, the EEG electrodes were placed in the head by the help of forward modelling of the OLB activity, and was found to reliably record the activity of the OLB. This approach was termed electrobulbogram (EBG). The EBG also followed response patterns demonstrated in non-human animal models. The manuscript brings up unique information and possibilities of recording activity from OLB humans that has not been possible this far. The manuscript is well written and the observations are well validated. Yet, there are many unclarities in the methods and analyses approach taken that should be clarified.

Major Edit:

R4 Major Comment 1:

Reviewer: One of the main validation in the study was the observation that ICL patient without OLB does not have gamma oscillatory activity similarly to healthy subjects. Figure 5b shows data for this one ICL patient. These data are difficult to compare with the corresponding data of healthy subjects presented in Figure 2 as this is an average over all subjects. It would be valuable to present single subject TFR data also to healthy participants for a comparison.

Reply: This is an interesting question that our data already partly answer. However, as suggested by the reviewer, we have assessed individual TFR maps in a similar manner as the one presented for the ICA patient, which is presented below (see Figure R1). Note though that this data can already be viewed by the readers from Figure 2F within the manuscript where the scatter plot shows the averaged TFR in the time-frequency of interest (indicated with a box within the result for the ICA patient). We also added a sentence in p. 6 in the first paragraph to clearly inform readers that they cannot infer the group level result to the individual participant. We are now stating:

“However, as Figure 2F demonstrate, an EBG response in the time-window of interest was not clearly detected in all individuals and it is our experience that the exact location (time/frequency) will differ slightly between individuals.”

Note that when viewing these TFR maps, EEG scalp-based recordings from humans differ in signal strength from intracranial recordings in animals and it is exceedingly rare to present individual data given the dependency on increased power by averaging across participants. Even when data is obtained from single electrodes directly recording from within olfactory cortex in humans, the signal is not very strong and variable (see Fig 4 in Zhou et al., 2019. *Nature Communication*, 19:1168).

Figure R1. Individual T-maps for normosmics (N=29).

R4 Major Comment 2:

Reviewer: The optimal location of the Electrobulbogram (EBG) channels were determined based on simulated leadfield. Please give more insight of what head models were used and how the leadfield was constructed.

Reply: The simulated leadfield was performed by assuming two dipoles in OB, radially laying in the same direction as the olfactory nerve. Four (4) co-centric spheres were used for the head model (which were the same as used in the main source reconstruction analyses). The dipoles were assumed to operate in gamma frequency and a forward model solution was used to calculate the leadfield. For more details of the underlying models and their assumptions, we refer the interested reader to the original paper (Cuffin and Cohen, 1979). This information can now be found in "Material and Method" section under subsection of "Electrode placement- Electrobulbogram". To highlight this information, we have emphasized where the interested reader can get a detailed understanding of the used method. The new text now reads:

"The dipoles momentum was assumed to face radially outward and the same head model (i.e. 4 co-centric sphere) as the main analysis was used to project the lead-field on the scalp level (Fig. 1B). The simulation suggested that the majority of the OB's energy concentrate on the forehead; therefore, optimal placement of the 4 electrodes were determined to be a curved configuration on the forehead slightly above the eye-brows, bilaterally, in addition to two mastoid electrodes as the reference electrodes (Fig. 1C; mastoid electrodes are not shown in the figure). For detailed implementation of lead-field estimation, please see (Cuffin and Cohen 1979)."

R4 Major Comment 3:

Reviewer: Source modelling was performed with DICS, which is a model based on suppression of coherent sources. Bilateral primary sensory areas including OLB are usually highly synchronous with each other and therefore the other of the sources is largely suppressed. Why was the DICS chosen for source-modelling these data as it can't give accurate reflection of neuronal activity in bilateral OLB. It should be shown that other source modelling approaches such as MNE or Loreta can be used to better localize distributed sources would give similar results.

Reply: Thank you for this insightful comment. However, we respectfully disagree that MNE source reconstruction is a good choice for this research question due to the step in the MNE pipeline where sources are projected back only to the cortical sheets and sources not on the cortical sheet are skewed and stretched for better fit. The OB is not included in MNE's cortical template. Moreover, MNE is commonly used for stable phase-locked events. However as we will discuss in more detail within the next comment (R4 Major Comment 4), due to inher-

ent variation in the human olfactory system, there is a weak phase-locking with stimulus onset. That said, we performed a new source reconstruction using eLORETA. As can be viewed in Figure R2, the result was similar but with a less focal map. This result is in line with a recent simulation study where it was demonstrated that DICS render more focal maps compared with eLORETA when assessing single source with low SNR (Halder et al., 2019). Due to the early timing of the synchronization we observed (<100ms), it is likely that there is only a single source given our time estimates of signal propagation through the olfactory system (see Supplementary Data, page 1). Because we are assessing a low common SNR in the gamma frequency, we argue that DICS is the most suitable source reconstruction algorithm for this dataset.

Figure R2. Signal source reconstruction using eLORETA. Yellow circle indicates location of the OB.

R4 Major Comment 4:

Reviewer: Only oscillatory power changes were analysed. Yet, the early responses are also phase-locked to the stimulus onset. This so called inter-trial phase locking is also known to play a significant role in sensory perception in other modalities. As OLB is in the first stages of olfactory processing, the gamma-band responses would be expected to be also phase-locked to the stimulus onset. This should be established.

Reply: This is an interesting concept that we had not previously assessed in our data and thank the reviewer for bringing this to our attention. We reanalyzed our data and found an inter-trial phase locking within the gamma-band in the range of our hypothesized time window of interest (see Supplementary Data for how we obtained this time window). We had a

significant phase-locking around 150ms after stimulus onset, a time that is slightly later than the TFR results (Figure 2 in paper). However, one should be aware of that, even though this is commonly assessed in the visual and auditory domain, to the best of our knowledge, no human odor scalp EEG study have assessed inter-trial phase locking. We believe that this is due to the fact that the human olfactory system inherently displays a much greater variability than the other systems due to variation in both stimulus presentation (olfactometer variability, odor travelling time, and flow-rate) and the underlying biological system (mucosa perfusion time and sniff intensity etc.) that other sensory systems do not suffer from. Due to this, even early responses are weakly phase locked to stimulus onset given its variability. As indicated in response to R4 Major Comment 1 above, even when data is obtained from single electrodes directly recording from within olfactory cortex in humans, phase-locking is not consistent across individual – in this case, evoked by an auditory signal with less variance than olfactory stimulation (see Fig 4 in Zhou et al., 2019. *Nature Communication*, 19:1168).

We have added and highlighted the results within the main text. The additional text on page reads:

“Early visual and auditory sensory responses are often characterized by a phase-locked response to stimulus onset (Lachaux et al., 1999). To assess whether we could detect stimulus phase-locking to odor onset in the obtained gamma band response in our EBG scalp recordings, we assessed a potential inter-trial phase locking effect. There was a change in phase-locked response to the onset of the odor stimuli, albeit at a slightly later time point than our temporal windows of interest used in analyses (~150ms), yet still within the hypothetical OB processing time window (see Supplementary data).”

R4 Major Comment 5:

Reviewer: In p. 10, N1 was reported to be present over parietal areas. I am confused as in the Method section, it is described that responses were recorded with 4 electrodes.

Reply: We agree with the reviewer that we were not very clear in our method on this account. In all experiments, we recorded conventional scalp EEG using either 64 (Study 1-2 & 4) or 32 (Study 3) scalp electrodes according to the traditional 10/20 placement. In addition to these, we applied 4 forehead electrodes (EBG) at the position where our leadfield analyses indicated strongest theoretical OB signal. Note, however, that our main analyses are based on the 4 EBG electrodes and not the complete EEG setup. We have clarified this information in the Material and Methods section under the subsection of “Electroencephalography, Electrobulbogram, and neuronavigation measurement” where we now state:

“In all studies, the EEG (acquired using either 32 or 64 electrodes, dependent on study) and EBG (acquired using 4 additional frontal electrodes) signal was sampled at 512 Hz using active electrodes (ActiveTwo, BioSemi, Amsterdam, The Netherlands) and band-pass

filtered at 0.01-100 Hz during recording within the ActiView software (BioSemi, Amsterdam, The Netherland)."

And in the last sentence of same paragraph we write:

"In Study 1 and 2, the EEG/EBG recording included 64 EEG scalp electrodes and 4 EBG electrodes. After the attachment of all electrodes, the positions of each electrode in stereotactic space were digitalized using an optical neuro-navigation system (BrainSight, Rogue Research, Montreal, Canada). /.../ In Study 3, data were recorded from 32 EEG scalp electrodes and 4 EBG electrodes and Study 4 used 64 EEG scalp electrodes and 4 EBG electrodes."

R4 Major Comment 6:

Reviewer: Also the only figure of this is figure showing the decay of the N1 component as function of trials. Please clarify and explain in detail the measurement and approach chosen.

Reply: *The manuscript does not clearly indicate that Figure 4 is not showing decaying N1 but the peak-to-peak magnitude of N1-P2/3 components over the Pz electrode – the measure that past studies demonstrate most closely tracks perceived odor intensity and habituation in scalp EEG measures of odors. In all experiments, we also recorded conventional EEG from the scalp (which can be found in Material and Method section and specifically addressed above). The signal in Pz were band-passed filter [1~30Hz] and ERPs were analyzed in which the local minima and local maxima in time interval (0~850ms) were detected as the peak on N1 and P2/3 respectively. The difference of these two peaks are plotted as function of trails in Figure 4C, showing that the cortical response is diminished over time, indicating that habituation is taking place.*

We added information to Figure 4's figure legend to more clearly indicate this.

"Fig 4. Lack of habituation of the EBG measure (N=21). A) Linear mixed model demonstrate that an odor habituation paradigm produce no significant change in power of the olfactory evoked synchronization across trials, slope = .008, $t(437) = 1.58$ $p > .11$ CI = [- .002 .02]). Blue circles in figure represent the first half and rust colored cubes the second half of the total number of trials. Unfilled circles and cubes represent individual values and dashed line indicate the slope as a function of trials. B) T-statistics with 1000 Monte Carlo permutations demonstrating no significant change in power ($p > .05$). Positive values indicate larger signal for early trials. Circles indicate individual values C) Pz potentials were band-pass filtered and the local minima and maxima in time interval of were detected as the peak of N1 and P2/3, respectively. Black curve shows the mean N1-P2/3 peak-to-peak responses in the olfactory event-related potentials (ERPs) over the Pz electrode as a function of trials and shaded blue area shows 95% confidence interval. The mean curve is smoothed for presentation purposes. Blue line represent the slope revealed

by the linear model and indicate a decrease in amplitude difference between N1-P2/P3 ERP components at electrode Pz across trials."

R4 Major Comment 7:

Reviewer: Also show the ERPs as function of time similar to that used for showing the TFR.

Reply: This is indeed a good comparison between data points and we thank the Reviewer for this suggestion. We have now added Supplementary Figure 5 to the supplementary data showing the ERPs as function of time for different trials. Delta (marked with black bar) shows the peak to peak magnitude that is shown in Figure 4C in the main manuscript. Note the diminishing difference over time for the N1-P2/3 difference over the Pz electrode; to date, the best neural correlate between human odor EEG scalp response and odor intensity.

"Supplementary Fig 5. Left panel indicate magnitude of ERP response (extracted from the Pz electrode) in color, displayed over time and trials. Right panel indicate mean ERP, extracted from the Pz electrode (parietal cortex), at trial 1, 15, 30, 45, and 60 across time. Blue dots show N1 peak and red dots show P2/3 peaks. The difference (Delta) is shown with black bar on the right side of plot."

R4 Major Comment 8:

Reviewer: Overall, I think that the impact of the study would be stronger if the present results from the OLB could be tied within the results and neuronal phenomena found at the later stages of cortical processing recorded with classical EEG recordings. More specifically, if

gamma oscillations characterize the processing in OLB, what are the next stages in processing the olfactory response? Perhaps the P50 or N1 responses.

Reply: We agree with the reviewer that this is an interesting question and also the next logical step. However, we respectfully disagree that this information should be included in the present manuscript. We argue this based on two aspects. First, the design is not optimized to assess this additional questions and we are currently conducting studies where we have optimized both the odor stimuli and the task performed by the participants to allow us to tease apart how the bulb interact with other odor areas in a systematic manner. Although we could use the current data to assess similar questions, it will be incomplete results that will not allow us to link change in neural data to change in either behavior or perception. Second, adding this research question to the manuscript would significantly expand the already extensive manuscript and transform it from a manuscript clearly describing a method to a paper that also address a question of function, albeit in a sub-par manner. Therefore, although we agree with the reviewer that the stated question is indeed interesting, we respectfully disagree that we should change the aims of the paper and instead address a different question than what the four experiments were designed to answer.

R4 Major Comment 9:

Reviewer: The lesion was suggested to control several physical factors, including respiration. Yet, it is quite clear that respiration and sniffing are intertwined in healthy participants. I am intrigued about how these would be coupled. Respiration rate is ~ 1 Hz while the odor perception was correlated with gamma-band oscillations. Could it be so that the gamma oscillations are cross-frequency coupled to the breathing rate?

Reply: Also this is a very interesting question that we are currently exploring in detail in a range of experiments. However, in these datasets, the onset was sniff-triggered which means that we cannot vary the sniff parameter in a systematic manner; either by modulating the onset of sniff cycle or sniff-to-odor magnitude manipulation. That said, as we are referring to above (R3 Comment 4), there is a bulb response seemingly occurring at the onset of the trial at a lower frequency than the EBG response. Given that the signal is sniff-locked, it seems likely that we can trigger an OB signal by sniffing alone, even in humans. Whether the gamma oscillations are cross-frequency coupled to sniffing is something that we are actively exploring in studies specifically design to assess and manipulate this signal. Therefore, although we agree with the reviewer that the stated question is indeed interesting, we respectfully disagree that we should change the aims of the paper and instead address a different question than what the four experiments were designed to answer. I realize that we now twice declined to address interesting research questions. However, given the fact that these studies were not optimized to address these new questions, we feel strongly about saving these for future experiments were we can make better conclusions from the resulting data.

We hope that the Reviewer agrees with this assessment. Nonetheless, we wish to thank the Reviewer for bringing these questions forward because it validates our own interests.

R4 Major Comment 10:

Reviewer: Please clarify, how the EEG channels were referenced.

Reply: Thanks for pointing out this unclear information. All data was always referenced to averaged Mastoidis (M1 and M2) electrodes except for source reconstruction. In our source reconstruction, we re-referenced to average of all electrodes. We emphasize this in p.18 in the section "Beamforming source reconstruction: localizing the OERS' cortical source" by the following section:

"To localize the cortical source of the detected OERS, EEG/EBG were re-referenced to average of electrodes and spectral density of the signal at the time period of 100 to 250 ms post-stimulus were estimated using fast Fourier algorithm with central frequency 60 Hz (i.e. the central frequency of the OERS) and taper smoothing parameter 5 Hz, meaning that the range from 55 Hz to 65 Hz were taken into the computation of cross spectral density for source localization (Fig. 1E)."

R4 Major Comment 11:

Reviewer: I find the summary confusing and not descriptive of the actual study. First, it does not mention that recordings are obtained from human subjects. Second, the description of EBG is quite misleading and it is used both as technique and result. Please clarify.

Reply: To be sure that it is clear that we record from the human olfactory bulb, we have added "human" to the main summary sentence that now reads (bold text is used here for highlighting the specific change):

*"We demonstrate that signals obtained via recordings from EEG electrodes at the nasal bridge represent responses from the **human** olfactory bulb - recordings we term Electrobulbogram (EBG)."*

However, with all due respect, we are not sure where the argued dichotomy between technique and results stem from. We argue that the technique itself described within the manuscript is the result. We speculate that it might be the very last sentence in the abstract that the Reviewer is referring to, where we indicate that this method could be used for clinical purposes. To make sure that readers do not believe that we are demonstrating its clinical use within the present manuscript, we have amended the sentence to now read (again, bold text used to emphasize changes):

*"The EBG will aid future olfactory-related translational work but can also **potentially** be implemented as an everyday clinical tool to detect pathology-related changes in human central olfactory processing in neurodegenerative diseases."*

Even though it is not clear to us what the Reviewer specifically means with this comment, we hope that our changes were able to sufficiently address Reviewer 4's concerns.

R4 Major Comment 12.

Reviewer: The introduction summarizes many prior studies of neuronal activity on OLB. It would be good if the also the seminal studies of the group by Gilles Laurent which reveal the computational functions and operations carried out by gamma oscillations in olfactory processing could be referred to (See for instance Laurent 2002 Nature Neuroscience Nov;3(11):884-95).

Reply: We wish to thank the Reviewer for pointing out the seminal work by Dr. Laurent that we indeed missed to cite. We are now citing this paper in our revision.

Minor Edit:

R4 Minor Comment 1:

Reviewer: Add minor tic marks to Fig 2 x-and y-axis.

Reply: Thank you for this suggestion. We have added the minor ticks to Figure2 and improved the figure further by using color map (Turbo) where color and underlying differences are perceived more linearly and true to data. Please see R1 Minor Comment 1 for more details.

R4 Minor Comment 2:

Reviewer: Add the N to the figure legends. The color-scale is difficult to read i.e. the different T-values are impossible to extract. Please change to a scale which gives more resolution for the values in the TFRs.

Reply: We agree that this is important information that will assist the reader when assessing the figures and have now added this to all figure legends, where applicable. As indicated above for comments R1 Minor Comment 1 and R4 Minor Comment 1, we have also changed the color scheme to better represent our underlying data.

Reviewers' Comments:

Reviewer #3:

Remarks to the Author:

The authors have responded in detail to each of the substantive critiques, with careful reasoning and improvements to the manuscript.

Reviewer #4:

Remarks to the Author:

The authors have revised the manuscript according to the reviewers' suggestions and the majority of my concerns have been answered. There is one major issue and some minor issues left that should be clarified.

Major

In the prior review, I asked for comparability of results between ICL patient single-subject data and group averaged data for healthy controls. The authors argue, correctly, that it is exceedingly rare to present single subject data for EEG recordings. Yet, they do it for the single-subject ICL patient data, which is rather contradictory to the prior argument. To convincingly establish that ICL patient does not have a gamma-band response to odor stimuli, additional analyses must be made. This evaluation can't be based on the visualization of the TFR alone now presented in Fig 5B since as presented for the rebuttal to the reviewers (Figure R1) individual TFRs are rather noisy. It should be shown that the strength of gamma band response to the ICL patient is above at least 2SDs of the confidence limit for the healthy controls.

Minor

Text p. 6 refers to that phase-locking to odor onset was assessed and it is in slightly later time-window than the gamma band response. Yet, this result is shown nowhere. Could you please add this result.

On p 2, it is written that using their power analysis that only 7 clean trials are needed to full statistical power. I assume the authors mean subjects as shown in supplementary figure 2? This result and number is quite unrealistic. What was the effect size used in the power analyses? I assume that authors have used rather large number to obtain this result.

I am confused about the Figure 1A. It shows that source modelling is performed after spectral and time frequency decomposition and after source modelling, there is head modelling stage. However, head models are used to create source models. I hope this is a mistake just in the visualization.

I would suggest adding the results from LORETA source estimation to the Supplementary material as this article is centrally dependent on the accuracy of the source model.

Response to reviewers: Non-invasive recording from the human olfactory bulb.

We thank the Editor and Reviewer #4 for their thoughtful comments on our manuscript. Below, we have addressed all comments in a point-by-point fashion. In summary, we have clarified some misunderstandings and made the text clearer and added a figure to the Supplementary material. All changes within the manuscript itself is highlighted with yellow marking. Below, our responses to reviewer comments are marked with blue text below and excerpts from the changed text within the manuscript are marked with *italic text*.

Reviewer #4 (Remarks to the Author):

R4. Major Comment1: In the prior review, I asked for comparability of results between ICL patient single-subject data and group averaged data for healthy controls. The authors argue, correctly, that it is exceedingly rare to present single subject data for EEG recordings. Yet, they do it for the single-subject ICL patient data, which is rather contradictory to the prior argument. To convincingly establish that ICL patient does not have a gamma-band response to odor stimuli, additional analyses must be made. This evaluation can't be based on the visualization of the TFR alone now presented in Fig 5B since as presented for the rebuttal to the reviewers (Figure R1) individual TFRs are rather noisy. It should be shown that the strength of gamma band response to the ICL patient is above at least 2SDs of the confidence limit for the healthy controls.

Reply: We thank the review for bringing up this to our attention. After reading over the manuscript again in detail, we realize that it was not clearly detailed in the manuscript that we had already presented the type of analyses that Reviewer #4 is asking for (Figure 5C). We have therefore added explanatory text to more clearly highlight what we did and what the result was. In sum, what our analyses demonstrate is that the effect is indeed more than 2SD (>2.5SD) away from the healthy controls in the predicted direction. On page 12-13, we are now stating:

“Using an identical experimental protocol as Study 1, we demonstrate that the ICA subject did not exhibit an EBG response following odor stimulation. Specifically, within the time and frequency window of interest, no significant EBG signal was observed for Odors compared with Air condition, Figure 5B. Visually assessing a single participant’s TRF result is inherently difficult due to its noisy structure. We therefore compared the strength of the signal in the time-window of interest to that of all participants in Study 1. As predicted, the mean EBG signal was 2.5SD below that of the mean of all participants in Study 1 (Figure 5C).”

Minor Edit

R4. Minor.Comment.1:

Reviewer: Text p. 6 refers to that phase-locking to odor onset was assessed and it is in slightly later time-window than the gamma band response. Yet, this result is shown nowhere. Could you please add this result.

Reply: As we outline in the previous response to reviewers, phase-locking to odors is inherently difficult to achieve in humans given that each odor arrives at slightly different times due to difference in volatility and mucosa perfusion as well as some minor differences in individual sniff strengths. To

optimize odor studies where phase-locking is assessed, odors with similar chemical structure as well as an olfactometer where very short tubing is used would be needed to obtain high quality data. Because our study was not design to assess this phenomenon and it is not a question of interest for the aims of the manuscript, we argue that additional figures are not needed in the manuscript to highlight the results. All that said, we do understand that we cannot expect that the Reviewer should blindly trust our word. The raw result of the odor vs air phase-locking data can therefore be viewed below (Figure R2-1).

Figure R2-1. Result for Phase-locked to onset. Odor is depicted with red color and clean air with green color. Pink area denote area of the 95% confidence interval.

However, after discussing this further, we came to the realization that this analysis cannot be directly linked to our analysis in the paper because the analyses is an aggregated view over all frequencies and interaction effects with the sniff is not contrasted out. Therefore, we further assessed the phase-locking response only within the gamma band and contrasting Odor and Air. Here, we the phase-locking response was equivalent to the finding in our main analysis presented in the manuscript (Figure R2-2). We have therefore amended the text in the manuscript to reflect this. Please see below for text extract.

Figure R2-2. Phase locked response for gamma power. The blue curve shows the phase locked gamma power for Odor-Air contrast and shaded area shows the 95% confidence interval. Around 100ms the phase locked gamma power of odor is significantly more than Air.

Hence, because the issue of phase-locking is outside the aim of the paper, the results just brings additional support to our main conclusion, and we already have many figures in a relatively straight-forward method paper, we argue that it is of only marginal benefit for the reader to include a figure to show the results beyond what we are now writing in the main text, namely:

“Early visual and auditory sensory responses are often characterized by a phase-locked response to stimulus onset. To assess whether we could detect stimulus phase-locking to odor onset in the obtained gamma band response in our EBG scalp recordings, we assessed a potential inter-trial phase locking effect in the gamma band and within the same contrast between Odor and Air. There was a change in phase-locked response to the onset of the odor stimuli around the same time point as in our temporal windows of interest used in our analyses (~100ms), thus bringing additional support to the notion that the EBG signal is an odor evoked response.”

If the Editor and Reviewer argues that a figure should be included in the manuscript, we will, of course, agree to this and in that case would favor inserting Figure R2-2 below in the supplementary material given its match with other analyses throughout the manuscript.

R4. Minor.Comment.2:

Reviewer: On. p 2, it is written that using their power analysis that only 7 clean trial are needed to full statistical power. I assume the authors mean subjects as shown in supplementary figure 2? This results and number is quite unrealistic. What was the effect size used in the power analyses? I assume that authors have used rather large number to obtain this result.

Reply: Thank you for observing this. We have now we changed the trial to subjects. To clarify, the effect size for this analysis was considered to be a z-value of 2.15 (p-value ~ 0.015), a target value

that we believe is a conservative assessment for future studies to aim towards. To put this in perspective, the assumed distribution is illustrated in Figure R2-3 for an $n=7$.

Figure R2-3. Assumed distribution for power analysis. H_1 is normal distribution with $\mu = .24$ and $\text{std} = .12$.

On page 6, we are now stating:

“In Study 1, an average of 52 % of all trials was removed from analyses due to artifacts. Thus, to determine the amount of data needed to detect a reliable signal from the EBG with the same statistical power as demonstrated in Study 1, individuals were stepwise added to power analyses. Only 7 artifacts free individuals were required to reach full statistical power (Supplementary Figure 2).”

R4. Minor.Comment.3:

Reviewer: I am confused about the Figure 1A. It shows that source modelling is performed after spectral and time frequency decomposition and after source modelling, there is head modelling stage. However, head models are used to create source models. I hope this is a mistake just in the visualization.

Reply: Thank you for pointing out this confusion in the reviewed figure. The figure is now improved to address this in a clearer way. Source and head models are independent from each other but both are dependent on the MNI anatomical T1 weight image. This was not clear in the previous figure. The new figure where this has been clarified can be seen below.

Figure 1. Overview of the methodological procedure to extract signal from olfactory bulbs. **A)** Flowchart of the procedures. **B)** A lead field simulation of olfactory bulb activity projected on the scalp using a symmetrically located dipole in each olfactory bulb (left/right). **C)** Electrode placement for the electrobulbogram (EBG) on the forehead and exemplary recordings. **D)** Multi-taper time frequency decomposition using two Slepian tapers. **E)** Cross-spectral density between scalp electrodes and EBG channels. **F)** Four concentric spheres used to construct the head model. **G)** The undetermined source model of every voxel of brain with gray matter probability more than 40% together with the digitalized sensor position of each individual and head model fed into dynamical imaging of coherent source to localize the cortical sources.

R4. Minor.Comment.2:

Reviewer: I would suggest adding the results from LORETA source estimation to the Supplementary material as this article is centrally dependent on the accuracy of the source model.

Reply: We have now added the eLORETA source estimates to the Supplementary material and are mentioning this in the manuscript on page 7.

Reviewers' Comments:

Reviewer #4:

Remarks to the Author:

Authors have now adequately addressed all concerns that I had.